# Self-Rewarding PPO: Aligning Large Language Models with Demonstrations Only

**Qingru Zhang**[†*] **Liang Qiu,**[◇] **Ilgee Hong**[†] **, Zhenghao Xu**[†] **, Tianyi Liu**[◇] **, Shiyang Li**[◇] **,
Rongzhi Zhang**[†] **, Zheng Li**[◇] **, Lihong Li**[◇] **, Bing Yin**[◇] **, Chao Zhang**[†] **, Jianshu Chen**[◇] **,
Haoming Jiang**[◇] **, Tuo Zhao**[†]
[†]Georgia Institute of Technology  [◇]Amazon
{qingru.zhang,tourzhao}@gatech.edu

## Abstract

Supervised fine-tuning (SFT) has emerged as a crucial method for aligning large language models (LLMs) with human-annotated demonstrations. However, SFT, being an off-policy approach similar to behavior cloning, often struggles with overfitting and poor out-of-domain generalization, especially in limited-data scenarios. To address these limitations, we propose Self-Rewarding PPO, a novel fine-tuning method that leverages on-policy techniques to enhance generalization performance. Our approach combines the strengths of SFT and proximal policy optimization (PPO) to achieve more effective alignment from demonstration data. At its core is a reward function designed as the log policy ratio between the SFT model and the pretrained base model. This function serves as an implicit reward signal, using the pretrained policy as a baseline and the SFT policy as a target. By doing so, it enables on-policy fine-tuning without relying on human preference annotations. The integration of this self-rewarding mechanism with PPO addresses key limitations of SFT, improving generalization, data efficiency, and robustness. Our empirical evaluation across a range of natural language processing tasks demonstrates that Self-Rewarding PPO consistently outperforms traditional SFT methods. The results highlight the effectiveness of our approach in aligning LLMs using demonstration data, particularly in scenarios where high-quality annotated data is scarce.

## 1 Introduction

Large language models (LLMs) exhibit remarkable performance in various tasks ranging from text generation to complex reasoning (e.g., Brown et al., 2020; Touvron et al., 2023; OpenAI, 2023). Their ability to generate coherent and contextually relevant text has enabled breakthroughs in areas such as creative writing, code generation, and conversational AI (Chen et al., 2021; Thoppilan et al., 2022; Bubeck et al., 2023; Anil et al., 2023). However, aligning these models to ensure both safety and utility remains a critical challenge.

Alignment refers to shaping model behavior to adhere to human values while avoiding harmful, biased, or unhelpful outputs (Bai et al., 2022a; Ganguli et al., 2022). The alignment process typically consists of two stages: (i) supervised fine-tuning on demonstration data, where models are fine-tuned on pairs of prompts and responses generated by experts (human or AI) to mimic desired behaviors (Wei et al., 2021; Chung et al., 2022; Zhou et al., 2023a; Tunstall et al., 2023); and (ii) preference learning, where preference data is used to learn a reward model, which is in turn used by a reinforcement learning (RL) step to fine-tune the model (Christiano et al., 2017; Ouyang et al., 2022; Stiennon et al., 2020; Bai et al., 2022b). In this work, we focus on the first stage: aligning language models from demonstration data without relying on preference annotations.

Supervised fine-tuning (SFT) has become the de-facto approach for learning desired behaviors from human-annotated demonstrations. This objective aligns closely with imitation

---

[*]Work completed during Qingru Zhang's internship at Amazon.

learning (Hussein et al., 2017; Osa et al., 2018). By maximizing the likelihood of expert's behaviors shown in the demonstration data, SFT is equivalent to performing behavior cloning, where the model aims to mimic the demonstrated actions (Bratko et al., 1995; Torabi et al., 2018; Florence et al., 2022). However, as an off-policy approach akin to behavior cloning, SFT typically relies on substantial volumes of high-quality data to achieve robust performance (Dubey et al., 2024). In the scenarios of limited data, SFT often suffers from overfitting to the training distribution, particularly with prolonged training, which hurts the model generalization on unseen examples and domains (Zhang et al., 2024; Chen et al., 2024).

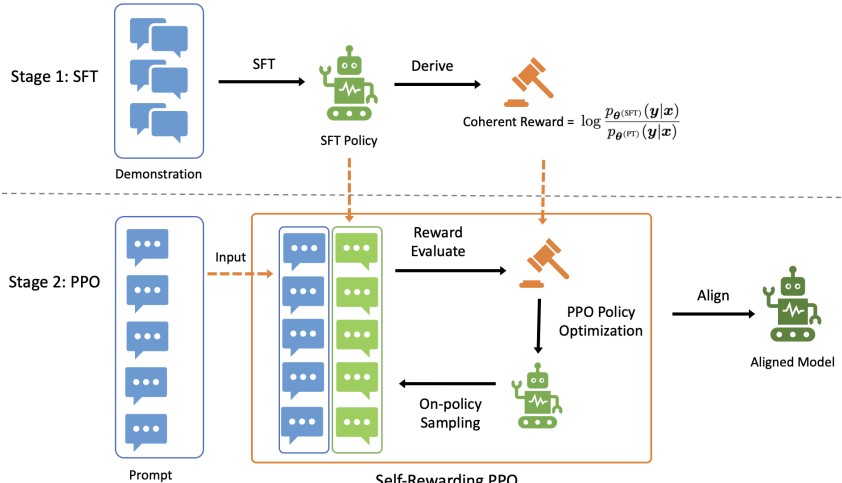

Figure 1: Illustration of Self-Rewarding Proximal Policy Optimization (SRPPO). SRPPO defines a reward based on the log-density ratio between the supervised fine-tuned (SFT) and pretrained policies. It then applies the reward to further fine-tune the SFT policy by proximal policy optimization. SRPPO leverages on-policy data to improve the out-of-distribution generalization of the learned policy.

Notably, on-policy training methods, such as proximal policy optimization (PPO, Schulman et al. (2017)), have been widely adopted in preference learning stage (Ouyang et al., 2022; Bai et al., 2022b). Unlike SFT, PPO generates diverse on-policy samples for fine-tuning. It enhances data diversity and makes training process adapt to model's evolving behaviors, improving generalization performance. Their success on preference learning motivates us to explore if on-policy training techniques can be leveraged to benefit SFT. However, a key challenges lies in deriving a meaningful reward signal from the data to enable on-policy training. A few recent studies attempt to address this challenge from two perspectives.

On one hand, inspired by advances in imitation learning, Li et al. (2024) and Sun & van der Schaar (2024) employ inverse reinforcement learning (IRL) (Ziebart et al., 2008; Ho & Ermon, 2016; Ghasemipour et al., 2020), to explicitly learn a reward model. While effective, these methods require training both the policy and reward models using a bi-level optimization framework, introducing significant complexity in terms of convergence and training stability.

On the other hand, SPIN (Chen et al., 2024) bypasses training an explicit reward model by assuming that the responses in demonstrations are always preferred over the model's on-policy samples. It applies DPO (Rafailov et al., 2024b) to fine-tune the model. As it does not involve an explicit reward, SPIN cannot accommodate additional prompts beyond those in the demonstrations during the training. Meanwhile, the preference assumption does not always hold, potentially leading to performance degradation. Despite these efforts, the design of meaningful reward signals from demonstrations remains largely unexplored and challenging.

In this study, we propose *Self-Rewarding PPO* (*SRPPO*), a novel fine-tuning method that bridges the gap between supervised fine-tuning and reinforcement learning fine-tuning, achieving more effective and robust alignment from demonstration through on-policy

training. At the core of our method, we propose a reward function named *coherent reward* that is designed as the log policy ratio between the supervised fine-tuning model (SFT policy) and the pretrained base model (pretrained policy). Specifically, our approach consists of two steps. First, we perform SFT to fine-tune the pretrianed base model using high-quality demonstrations – pairs of prompts and desired responses. Next, we derive the coherent reward from the SFT and pretrained policies. Guided by the coherent reward, we apply PPO to continuously fine-tune the model using a set of prompts. Figure 1 illustrates the two stages of our method.

Unlike existing IRL methods Li et al. (2024), SRPPO eliminates the need to train a reward model. Instead, the coherent reward is derived directly from the SFT policy, which is the same model to be trained, offering a simple but effective self-rewarding mechanism. This reward design is inspired by *coherent soft imitation learning* (Watson et al., 2024). Coherent reward leverages the pretrained policy as a baseline and the SFT policy as a target. It establishes a training direction that pushes the model's behavior from the pretrained baseline to the mid-aligned SFT policy. The PPO stage then refines the model further along this alignment direction with on-policy sampling. Moreover, compared to SPIN, SRPPO allows the use of additional prompts beyond those in the demonstration data during the PPO step. This flexibility is particularly advantageous in scenarios where high-quality responses are scarce, but obtaining abundant similar prompts is relatively easy. In such cases, a small amount of high-quality demonstration data can be used during SFT to establish an alignment direction that is captured by the coherent reward. Subsequently, during the PPO fine-tuning step, additional prompts can be utilized to sample more on-policy responses, which are then evaluated by the coherent reward, further refining the model along the established direction. Empirically, we observe that the coherent reward generalizes effectively from a small set of representative demonstrations to a broader range of prompts (Section 4). Therefore, Self-Rewarding PPO not only augments training data with on-policy samples, but also allows the use of additional prompts to enhance alignment.

We conduct experiments to demonstrate the effectiveness of Self-Rewarding PPO using LLAMA3-8B and Mistral-7B as our base model. Empirical results show that SRPPO significantly enhances fine-tuning performance compared to SFT and other alternative methods across various evaluation benchmarks, demonstrating its effectiveness in improving model alignment and generalization.

## 2 Background

Consider a language model parameterized by $\boldsymbol{\theta}$ and denote its output probability (or policy) by $p_{\boldsymbol{\theta}}(\boldsymbol{y}|\boldsymbol{x})$, where $\boldsymbol{x} = [x_1, \ldots, x_n]$ is the sequence of input prompts and $\boldsymbol{y} = [y_1, \ldots, y_m]$ is the sequence of output responses. LLMs are typically auto-regressive models: they generate tokens one-by-one, and predict the output probability of $y_j$ given tokens in $\boldsymbol{x}$ and $\boldsymbol{y}_{<j} = [y_1, \ldots, y_{j-1}]$ ($y_{<1}$ is null):

$$p_{\boldsymbol{\theta}}(\boldsymbol{y}|\boldsymbol{x}) = \prod_{j=1}^{m} p_{\boldsymbol{\theta}}(y_j|\boldsymbol{x}, \boldsymbol{y}_{<j}).$$

This process constitutes a Markov decision process (MDP), where the state transitions are deterministic and the model generates tokens sequentially at every given position, leveraging only the sequence of previous tokens. In the following, we discuss two common procedures for fine-tuning $\boldsymbol{\theta}$: (i) supervised fine-tuning (SFT) over a demonstration dataset, and (ii) reinforcement learning with human feedback (RLHF) over a preference dataset.

**SFT.** Supervised fine-tuning (SFT) aligns or adapts a pre-trained LLM to specific tasks, (e.g., instruction following, code generation, math reasoning). This process relies on a demonstration dataset $\mathcal{D} = \{(\boldsymbol{x}, \boldsymbol{y})\}$ that comprises prompts $\boldsymbol{x}$ sampled from the task distribution $\rho$, and their responses $\boldsymbol{y}$ annotated by experts $p_{\text{expert}}(\cdot|\boldsymbol{x})$. SFT uses a maximum-likelihood objective:

$$\max_{\boldsymbol{\theta}} \ell_{\text{SFT}}(\boldsymbol{\theta}) = \mathbb{E}_{(\boldsymbol{x}, \boldsymbol{y}) \sim \mathcal{D}} \left[ \log p_{\boldsymbol{\theta}}(\boldsymbol{y}|\boldsymbol{x}) \right]. \tag{1}$$

Clearly, the above problem shares the same optimal solution as $\min_{\boldsymbol{\theta}} \mathbb{E}_{\boldsymbol{x} \sim \rho}[D_{\text{KL}}(p_{\text{expert}}(\cdot|\boldsymbol{x}) \| p_{\boldsymbol{\theta}}(\cdot|\boldsymbol{x}))]$. $\ell_{\text{SFT}}(\boldsymbol{\theta})$ attains its optimum when the model

$p_{\theta}$ aligns perfectly with the expert behavior. So the fine-tuned model is expected to generate responses that resemble those of the expert. Therefore, SFT is closely related to imitation learning Osa et al. (2018), whose goal is to mimic the policy of an expert.

**RLHF.** RL fine-tuning over a preference dataset is the second stage of aligning LLMs, after the SFT stage. Suppose we have a deterministic reward model $r(x, y)$ that evaluates a given prompt-response pair $(x, y)$. RLHF fine-tunes the model by solving the following RL problem:

$$
\begin{aligned}
\max_{\theta} \ell_{\text{RL}}(\theta) = {} & \mathbb{E}_{x \sim \rho, y \sim p_{\theta}(\cdot|x)} \left[ r(x, y) \right] \\
& - \lambda \mathbb{E}_{x \sim \rho} \left[ D_{\text{KL}}(p_{\theta}(\cdot|x) \| p_{\text{ref}}(\cdot \| x)) \right],
\end{aligned}
\tag{2}
$$

where $p_{\text{ref}}$ is a reference model. Due to the intractability of computing the KL regularization over all possible outputs $y$, (2) is typically solved by policy optimization techniques such as REINFORCE (Williams, 1992; Ahmadian et al., 2024) and PPO (Schulman et al., 2017).

To obtain a reward model $r(x, y)$, RLHF often assumes a preference dataset $\mathcal{M} = \{x, y_w, y_l\}$, where each data contains a pair of output $(y_w, y_l)$ for prompt $x$. Here, $y_w$ is preferred over $y_l$ by human annotator, denoted as $y_w \succ y_l$ (Christiano et al., 2017; Ouyang et al., 2022). The Bradley-Terry model (Bradley & Terry, 1952) is used to model the probability of choosing $y_w$ over $y_l$:

$$
\mathbb{P}(y_w \succ y_l | x) = \sigma(r(x, y_w) - r(x, y_l)),
$$

where $\sigma(\cdot)$ is the sigmoid function. The reward model is trained with the following objective:

$$
\max_{r(\cdot, \cdot)} \ell_{\text{RM}} = \mathbb{E}_{(x, y_w, y_l) \sim \mathcal{M}} \left[ \log \left( \sigma \left( r(x, y_w) - r(x, y_l) \right) \right) \right].
$$

It is widely shown that models trained by episodically learning the policy (2) and learning the reward often outperforms those that are only trained using SFT (Ouyang et al., 2022). The reward model guides the performance of the LLM and allows a better generalization ability due to incorporating additional preference data from human annotator.

**Discussion.** In this study, we focus on aligning LLMs using demonstration data. As shown in (1), SFT is an off-policy approach akin to behavior cloning, where the model is fine-tuned to mimic expert behavior solely based on the provided data. Consequently, it often suffers from overfitting to training distribution, leading to subpar out-of-domain generalization. In contrast, RL fine-tuning in (2) is an on-policy method that samples responses directly from the model's current policy and optimizes it to maximize the reward of subsequent samples. This adaptability allows the training process to align with the model's evolving behavior, resulting in improved generalization and robustness. Motivated by this, the paper studies the following question:

*Can we leverage on-policy training techniques to bridge the gap between SFT and RL fine-tuning, thereby enhancing alignment from demonstrations only?*

In next section, we delve into this prospect and address this question with our solution.

## 3 Method

Our proposed method, Self-Rewarding PPO (SRPPO), combines the strengths of both SFT and RL fine-tuning. At the core of SRPPO, we introduce a novel reward function, *Coherent Reward*, which establishes an alignment direction coherent with supervised fine-tuning stage, thereby enabling continuous refinement through RL fine-tuning.

### 3.1 Coherent Reward

To enable on-policy training using demonstration data, we propose Coherent Reward, a novel reward function derived as the log policy ratio between the initial pretrained model (pretrained policy $p_{\theta^{(\text{PT})}}$) and the model fine-tuned on demonstrations (SFT policy $p_{\theta^{(\text{SFT})}}$). Specifically, for any pair $(x, y)$, the coherent reward is defined as

$$
\tilde{r}(x, y) = \log \frac{p_{\theta^{(\text{SFT})}}(y|x)}{p_{\theta^{(\text{PT})}}(y|x)} = \log \frac{\prod_{j=1}^{m} p_{\theta^{(\text{SFT})}}(y_j|x, y_{<j})}{\prod_{j=1}^{m} p_{\theta^{(\text{PT})}}(y_j|x, y_{<j})}.
\tag{3}
$$

Our coherent reward is inspired by the coherent soft imitation learning method Watson et al. (2024). Intuitively, it leverages the pretrained policy as a baseline and the SFT policy as a target. For a pair of prompt and response, it quantifies the divergence between two policy on this pair, thereby establishing a training direction that transitions the model's behavior from the pretrained baseline to the mid-aligned SFT policy. For a given prompt-response pair, the reward quantifies the divergence between these two policies on the pair, thereby establishing a training direction that transitions the model's behavior from the pretrained baseline to the mid-aligned SFT policy. We then leverage this reward for subsequent RL fine-tuning, ensuring that the RL fine-tuning effectively builds upon the improvements of SFT stage, and further refine the model along this alignment trajectory.

## 3.2 Self-Rewarding PPO

As illustrated in Figure 1, our Self-Rewarding PPO method consists of two sequential training stages:

1. **Supervised fine-tuning**: Given a demonstration dataset $\mathcal{D} = \{(x, y)_i\}_i$, we fine-tune a pretrained base model $p_{\theta(\text{PT})}$ on $\mathcal{D}$ by optimizing (1), and obtain the SFT policy $p_{\theta(\text{SFT})}$, from which we derive the coherent reward $\tilde{r}$ as in (3).

2. **RL fine-tuning**: Given a prompt set $\mathcal{P} = \{y_i\}_i$, we further perform the on-policy RL fine-tuning to continuously refine the SFT policy by optimizing the objective (2), where the reward value is our coherent reward.

For the RL fine-tuning stage, we use PPO as the policy optimization algorithm. Please see Appendix B for the details of PPO. Notably, other algorithms such as REINFORCE (Williams, 1992), GRPO (Shao et al., 2024), or RLOO (Ahmadian et al., 2024) can also be applied as alternatives.

When employing PPO to fine-tune the model, we treat states containing an [EOS] token as absorbing states. We assign the coherent reward at the process level as defined in (4). Alternatively, we can revise the process-level coherent reward to a token-wise reward $r(y_j|x, y_{<j}) = \log \frac{p_{\theta(\text{SFT})}(y_j|x, y_{<j})}{p_{\theta(\text{PT})}(y_j|x, y_{<j})}$ and assign it at token level, which we discuss further in Appendix E.

$$r(y_j|x, y_{<j}) = \begin{cases} \tilde{r}(x, y) & \text{if } y_j = [\text{EOS}] \text{ or } j = m, \\ 0 & \text{otherwise.} \end{cases} \tag{4}$$

Our coherent reward is straightforward yet meaningful. It offers a simple and effective self-rewarding mechanism for on-policy alignment training from demonstrations, without requiring rewarding learning or inverse reinforcement learning. Its advantages mainly stems from two key perspectives regarding on-policy responses $y \sim p_\theta(\cdot|x)$ and input prompts $x \sim \rho$. First, the coherent reward is derived from the same model being fine-tuned (i.e., SFT policy). Notably, during RL fine-tuning, on-policy responses $y$ are sampled from the same model. Consequently, the coherent reward becomes inherently sensitive to variations in the model's own responses $y \sim p_\theta(\cdot|x)$. Compared to independent reward models, the coherent reward can capture subtle changes in $y$, providing more accurate and adaptive reward evaluations. Second, our method enables the inclusion of additional prompts beyond those in the demonstration dataset. This is a substantial advantage compared to methods like SPIN (Chen et al., 2024) that rely on DPO and are limited to demonstration prompts. This flexibility can be particularly beneficial when high-quality, human-annotated responses are scarce but similar prompts can be easily obtained. In such cases, SFT is prone to overfitting. In contrast, SRPPO allows us to first fine-tune the model with a small amount of demonstration data to establish an alignment direction. Subsequently, we can sample more prompts from task distribution $x \sim \rho$, and utilize them and their on-policy samples to continue refining the model along the alignment direction given by the coherent reward. In Section 4, we empirically show how the coherent reward generalizes effectively from a small amount of demonstration data to a broader range of prompts. We summarize Self-Rewarding PPO in Algorithm 1.

Table 1: Fine-tuning results of Mistral-7B under the minimum overlap setup. SFT is conducted with Tulu-v2 only. The best results are shown in **bold**. The average score is calculated by first averaging two scores of each tasks, and taking averaging of them across four tasks.

| Method | IFeval L.Acc / S.Acc | GSM8k EM | GPQA CoT EM / Non-CoT EM | AlpacaEval LC win rate / Win rate | All Ave. |
|---|---|---|---|---|---|
| Pretrain Baseline | 30.58 / 29.38 | 37.3 | 12.50 / 27.23 | 0.07 / 0.12 | 21.81 |
| SFT | 42.45 / 40.53 | 46.47 | 23.88 / 26.34 | 8.95 / 4.60 | 29.96 |
| SFT (Extended) | 39.21 / 35.49 | 29.04 | 16.74 / **29.46** | 9.75 / 4.97 | 24.22 |
| SPIN | 45.08 / 38.73 | 42.99 | 19.87 / 26.56 | 5.81 / 4.29 | 28.29 |
| **SRPPO** | **47.60 / 41.37** | **46.93** | **24.33** / 26.56 | **12.47 / 13.23** | **32.43** |

Table 2: Fine-tuning results of LLAMA3-8B under the minimum overlap setup. SFT is conducted with Tulu-v2 only. The best results are shown in **bold**.

| Method | IFeval L.Acc / S.Acc | GSM8k EM | GPQA CoT EM / Non-CoT EM | AlpacaEval LC WR / WR | All Ave. |
|---|---|---|---|---|---|
| Pretrain Baseline | 28.30 / 26.85 | 50.11 | 12.95 / 27.23 | 0.23 / 0.25 | 24.50 |
| SFT | 28.42 / 28.30 | 47.69 | **25.67** / 29.69 | 9.48 / 4.97 | 27.74 |
| SFT (Extended) | 34.77 / 32.13 | 45.19 | 16.74 / **32.14** | 10.41 / 5.34 | 27.74 |
| PPO w/ a preference RM | 31.06 / 30.22 | 48.90 | - / - | - / - | - |
| **SRPPO** | **41.49 / 37.41** | **51.10** | 18.30 / 30.13 | **11.86 / 7.95** | **31.17** |

## 4 Experiments

We evaluate the effectiveness of Self-Rewarding PPO by fine-tuning the pretrained Mistral-7B (Jiang et al., 2023) and LLAMA3-8B (Dubey et al., 2024) models. The evaluation covers a diverse set of benchmarks, including instruction following (IFEval, Zhou et al. (2023b)), math reasoning (GSM8k, Cobbe et al. (2021)), graduate-level question answering (GPQA, Rein et al. (2023)), and conversational ability (AlpacaEval, Dubois et al. (2024)). Our experiments highlight the following advantages of SRPPO:

• **Improved performance without additional human annotations**: Without introducing new human-annotated data, SRPPO significantly enhances model performance across a wide range of evaluation benchmarks compared to SFT and other alternatives.

• **Effective generalization to additional prompts**: SRPPO enables the inclusion of additional prompts for RL fine-tuning, thus further improves model performance. This demonstrates that the coherent reward can effectively generalize from human-annotated demonstration data to a broader range of prompts, enhancing alignment without new annotations.

### 4.1 Experimental Setup

**Models and Datasets.** We adopt the pretrained Mistral-7B (Jiang et al., 2023) and LLAMA3-8B (Dubey et al., 2024) as our base models, and then conduct the supervised and RL fine-tuning to evaluate our method. For training, we leverage two datasets: TULU-v2-mix (Ivison et al., 2023) and UltraFeedback (Cui et al., 2024). TULU-v2-mix is a mixed collection of high-quality instruction datasets, comprising 326k examples from 11 diverse sources. UltraFeedback is a large-scale, fine-grained preference dataset containing 64k examples that are realted to aspects of instruction-following, truthfulness, honesty, and helpfulness. Since TULU-v2-mix is a high-quality dataset known for consistently improving model capabilities across various tasks, we primarily use it for supervised fine-tuning, ensuring an initial alignment of the models. To evaluate the generalization of our coherent reward beyond the demonstration data, we use prompts from UltraFeedback during the PPO fine-tuning stage. This setup allows us to examine how well our reward mechanism transfers to new prompts without additional human annotations.

Since the coherent reward is derived from the SFT policy, the choice of SFT training data is crucial to its effectiveness. Empirically, we find that overlapping the SFT demonstration data with the PPO prompt data helps derive a more robust coherent reward. To systematically evaluate the generalization of our approach, we consider the following experimental setups for selecting SFT training data in SRPPO:

**1. Minimum overlap**: We conduct SFT only on TULU-v2-mix, ensuring minimal overlap between the SFT training pairs and PPO training prompts. This setup is designed to assess the generalization capability of SRPPO when the PPO stage encounters prompts not seen during SFT. Tables 1 and 2 presents results in this setting.

**2. Medium overlap**: To introduce a controlled degree of overlap, we sample 9k prompts from UltraFeedback and annotate them with high-quality responses generated by GPT-4. The models are first fine-tuned using TULU-v2-mix, followed by additional fine-tuning on this small subset of UltraFeedback demonstrations. Table 3 shows the results.

**3. Diminished overlap**: We first fine-tune the models using TULU-v2-mix to establish an initial alignment. We then conduct additional supervised fine-tuning using both the small UltraFeedback demonstration subset and an additional 40k examples from TULU-v2-mix to further refine the models. Table 4 presents the results of this setup.

These setups allow us to analyze how different levels of training data overlap influence the effectiveness of the coherent reward.

**Evaluation.** We evaluate model performance across different perspectives, including instruction following (IFEval, Zhou et al. (2023b)), math reasoning (GSM8k, Cobbe et al. (2021)), graduate-level question answering (GPQA, Rein et al. (2023)), and conversational ability (AlpacaEval, Dubois et al. (2024)). For IFEval, we report both instruct-level loose (L.Acc) and strict (S.Acc) accuracies. For GSM8k, we evaluate the 5-shot performance and report the exact match (EM). For GPQA, we assess both few-shot and few-shot Chain-of-Thought (CoT) performance (Wei et al., 2022b). These evaluations are conducted using the 'lm-evaluation-harness' framework (Gao et al., 2024) under its default settings. For AlpacaEval, we report length-controlled win-rate and overall win-rate.

**Implementation Details.** We use *PyTorch* (Paszke et al., 2019) to implement all the algorithms. Our implementation is based on the publicly available *Huggingface Transformers*[1] (Wolf et al., 2019) and OpenRLHF (Hu et al., 2024) code-base. All the experiments are conducted on NVIDIA A100 GPUs.

Regarding the hyperparameters of SFT, we set the batch size as 128 and trainig epochs as 2, choose the learning rates from $\{1 \times 10^{-5}, 5 \times 10^{-6}, 1 \times 10^{-6}, 5 \times 10^{-7}\}$, and pick the optimal learning rate for both SRPPO and baseline methods. For the hyperparameters of PPO, we set the rollout buffer size as 1024, the training batch size as 128, the KL coefficient as 0.2 or 0.5, and the clipping coefficient as 0.2. We initialize the critic model from the SFT policy, set its learning rate as $9 \times 10^{-6}$ and warmup the critic fine-tuning for 35 rollout buffers. We then fine-tune the actor model for 2 episodes and select the actor learning rates from $\{5 \times 10^{-8}, 2 \times 10^{-8}, 1 \times 10^{-8}\}$.

**Baselines.** We compare Self-Rewarding PPO with the following methods:

• *SFT*: It is the standard approach for aligning LLMs with demonstration data that optimizes (1). SFT is also the model from which SRPPO continues to fine-tune. We set the training epochs to 2, as this typically yields the best performance. Given different training data, we compare SRPPO against its SFT-only stage to showcase the effectiveness of PPO fine-tuning with our coherent reward.

• *SFT (Extended)*: This baseline extends the fine-tuning of the SFT policy by running additional SFT epochs. Specifically, it starts from the SFT policy and undergoes further fine-tuning, e.g., for a total of 6 epochs, to examine how prolonged SFT affects model performance.

---

[1]https://github.com/huggingface/transformers

Table 3: Fine-tuning results of Mistral-7B and LLAMA3-8B under the medium overlap setup. SFT is conducted with Tulu-v2 and a small number of demonstrations of Ultrafeedback. The best results are shown in **bold**.

| Model | Method | IFeval L.Acc / S.Acc | GSM8k EM | GPQA CoT EM / Direct EM | AlpacaEval LC WR / WR | All Ave. |
|---|---|---|---|---|---|---|
| **Mistral-7B** | Pretrain Baseline | 30.58 / 29.38 | 37.3 | 12.50 / 27.23 | 0.07 / 0.12 | 21.81 |
| | SFT w/ a subset | 46.40 / 43.05 | 35.18 | 18.75 / 25.45 | **22.63** / 16.21 | 30.36 |
| | **SRPPO** | **49.40 / 44.00** | **41.39** | **20.31 / 27.23** | 21.73 / **21.18** | **33.33** |
| **LLAMA3-8B** | Pretrained Baseline | 28.30 / 26.85 | 50.11 | 12.95 / 27.23 | 0.23 / 0.25 | 24.50 |
| | SFT w/ a subset | 31.53 / 30.58 | 47.31 | **20.31 / 32.59** | **19.72** / 14.29 | 30.46 |
| | **SRPPO** | **45.44 / 40.17** | **53.22** | 20.09 / 31.47 | 19.20 / **23.48** | **35.79** |

Table 4: Fine-tuning results of Mistral-7B under the diminished overlap setup. We first fine-tune the models using TULU-v2-mix, and then conduct additional SFT using both the small UltraFeedback demonstration subset and an additional 40k examples from TULU-v2-mix to refine the model.

| Method | IFeval L.Acc / S.Acc | GSM8k EM | GPQA CoT EM / Direct EM | AlpacaEval LC win rate / Win rate | All Ave. |
|---|---|---|---|---|---|
| Pretrain Baseline | 30.58 / 29.38 | 37.30 | 12.50 / 27.23 | 0.07 / 0.12 | 21.81 |
| SFT | 47.36 / 44.36 | 27.52 | 15.85 / 27.23 | 13.57 / 7.52 | 26.37 |
| **SRPPO** | **48.56 / 44.72** | **30.55** | **19.87 / 27.23** | **15.59 / 9.01** | **28.26** |

• *PPO with an independent preference reward model*: Instead of using our coherent reward, this baseline employs a publicly available preference reward model (Fsfairx-LLAMA3-RM; Dong et al., 2024) for PPO fine-tuning after the SFT stage. This baseline showcases the comparison between our coherent reward that does not rely on additional preference data, and a reward models trained from preference data.

• *SPIN* (Chen et al., 2024): This baseline is a self-play-based fine-tuning method where the same target LLM generates synthetic data and critiques its own outputs.

### 4.2 Main Results

We present our main results in Tables 1 and 2. This corresponds to the first setup of minimum overlapping for selecting SFT data, where we perform SFT only on TULU-v2-mix and then apply PPO using UltraFeedback prompts. As illustrated, SRPPO consistently outperforms baseline methods on IFEval, GSM8k and AlpacaEval, achieving the best overall average scores for both Mistral-7B and LLAMA3-8B. While SFT (Extended) achieves the best Direct EM on GPQA, SRPPO still shows competitive performance. Particularly, extending SFT training to more epochs improves performance in some domains, such as question answering and instruction following. However, prolonged SFT also leads to overfitting to the training distribution, negatively impacting out-of-domain generalization. As observed in Table 2, extending SFT on TULU-v2-mix to 6 epochs (compared to 2 epochs) improves accuracy on IFEval and GPQA, likely due to the fact that most TULU-v2-mix examples belong to similar domains. However, this extended training reduces performance on math reasoning (Tables 1 and 2). This result validates our argument in Section 1 that prolonged SFT tends to cause overfitting, thereby hurting generalization to out-of-domain tasks. In contrast, SRPPO enhances model generalization, yielding performance gains across all four tasks. As shown in Table 2, SRPPO significantly improves performance on both instruction following and math reasoning, as it effectively enables on-policy training on additional prompts. These results indicate that SRPPO can effectively leverage additional prompts to enhance model capabilities without preference annotations.

Furthermore, as seen in Table 2, PPO with an independent preference reward model provides only marginal improvements over SFT, whereas SRPPO consistently outperforms SFT across

all benchmarks. This suggests the effectiveness of our self-rewarding mechanism. Since the coherent reward is derived directly from the SFT policy, it is inherently sensitive to variations in the model's own responses $y \sim p_\theta$ during RL fine-tuning. Compared to independent reward models (Table 2), we hypothesize that the coherent reward can capture subtle changes in $y$, enabling more accurate and adaptive reward evaluations. Additionally, we compare SRPPO with SPIN in Table 1. Similar to SRPPO, SPIN is initialized from the SFT policy. However, unlike SRPPO, SPIN conducts its first-iteration DPO using the full 350k examples from TULU-v2-mix, a dataset significantly larger than the one used by SRPPO. Despite this data advantage, while SPIN improves upon SFT, SRPPO consistently outperforms SPIN, demonstrating its effectiveness as a fine-tuning method for alignment with demonstrations.

Tables 3 and 4 present the results for the setups of medium overlap and diminished overlap (c.f., Section 4.1), respectively. In Table 3, during the SFT stage, we first fine-tune the models on large-scale TULU-v2-mix to establish their basic capabilities, followed by additional refinement using a 9k subset of high-quality demonstrations, where prompts are sampled from UltraFeedback and responses are annotated by GPT-4. This results in a medium overlap between the SFT prompts and the PPO prompts. We observe that SFT with this 9k subset significantly improves instruction-following and conversational ability but severely degrades math reasoning due to the scarcity of math-related data in this subset, inducing overfitting to training domains. In contrast, SRPPO effectively mitigates this issue, recovering math reasoning performance and yielding a substantial 4.06% EM improvement for Mistral-7B, which is consistent with observations in Tables 1 and 2. More importantly, SRPPO further enhances IFEval accuracy, as the prompt overlap leads to a coherent reward that exhibits better generalization on PPO prompts. In the diminished-overlap setup, we begin with the SFT policy from the medium-overlap setup and further fine-tune it using 40k additional samples from TULU-v2-mix. This step reduces the overlapping effects introduced by the 9k UltraFeedback demonstration subset. Even under this setup, SRPPO consistently outperforms SFT, demonstrating that the coherent reward effectively generalizes to additional prompts. These results confirm that SRPPO enhances model performance through on-policy training with additional prompts, reinforcing its ability to improve generalization without relying on preference annotations.

## 5 Discussions

Our coherent reward approach can be applied beyond PPO, and to other on-policy training algorithms, including REINFORCE (Williams, 1992), RLOO (Ahmadian et al., 2024), GRPO (Shao et al., 2024), and VinePPO (Kazemnejad et al., 2024). Additionally, it has the potential to serve as an effective evaluator, facilitating tasks such as SFT data filtering by providing a principled method for assessing response quality, thereby reducing the need for human involvement in the loop.

Empirically, we observe that the effectiveness of the coherent reward can be influenced by both the SFT training quality and the generalization ability of the pretrained base model. If the pretrained model has limited generalization to unseen domains and out-of-distribution prompts, such as a small-size model like Phi-2 (Javaheripi et al., 2023), its coherent reward faces challenges to generalize on a board range of prompts. Additionally, the quality of the SFT data and SFT training process play a crucial role in determining the effectiveness of the coherent reward. If the demonstration data is of low quality or if the SFT stage suffers from overfitting, the derived coherent reward may exhibit reduced generalization capability on unseen prompts.

Moreover, we also explore modifying the coherent reward (3) into a token-wise reward. However, this approach introduces challenges related to length degeneration, complicating the training process. We provide a detailed discussion in Appendix E and leave this for future exploration. Due to space limit, we dicuss the related work in Appendix C.

## 6 Conclusions

In this study, we introduced Self-Rewarding PPO (SRPPO), a novel fine-tuning framework that bridges the gap between supervised fine-tuning (SFT) and reinforcement learning

(RL) fine-tuning by leveraging a coherent reward derived directly from the SFT policy. SRPPO enables on-policy fine-tuning using demonstration data alone, making it a scalable and effective approach for LLM alignment. Experimental results demonstrate that SRPPO achieves significant improvements over SFT methods and other alternatives across multiple benchmarks, showcasing its effectiveness.

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

# A    Detailed Algorithm of SRPPO

---

**Algorithm 1** SRPPO

---

1: **Input:** Demonstartion dataset $\mathcal{D}$; Prompt set $\mathcal{P}$; Pretrained LLM $p_{\boldsymbol{\theta}^{(\text{PT})}}$.
2: **Supervised fine-tune** $p_{\boldsymbol{\theta}^{(\text{PT})}}$ using $\mathcal{D}$ and obtain $p_{\boldsymbol{\theta}^{(\text{SFT})}}$.
3: **Derive** $\tilde{r}$ using $p_{\boldsymbol{\theta}^{(\text{PT})}}$ and $p_{\boldsymbol{\theta}^{(\text{SFT})}}$.
4: **PPO fine-tune** $p_{\boldsymbol{\theta}^{(\text{SFT})}}$ using $\mathcal{P}$ and the reward $\tilde{r}$.
5: **Output**: the aligned model.

---

# B    PPO Algorithm

This section describes the PPO algorithm.

---

**Algorithm 2** Proximal Policy Optimization (PPO)

---

**Input:** Initial actor model $\pi_{\theta_{\text{init}}}$, critic model $V_\phi$, reward function $r$, task prompts $\mathcal{D}$, clip range $\epsilon$, batch size $B$.

1: Initialize $\pi_\theta \leftarrow \pi_{\text{init}}$.
2: **for** iteration $= 1, 2, \ldots$ **do**
3:     Sample batch of prompts $\{\boldsymbol{x}_i\}_{i=1}^B \subseteq \mathcal{D}$.
4:     Generate responses $\{\boldsymbol{y}_i\}_{i=1}^B$ where $\boldsymbol{y}_i \sim \pi_\theta(\cdot|\boldsymbol{x}_i)$.
5:     Initialize batch data $\mathcal{B} = \varnothing$.
6:     **for** index $= 1, 2, \ldots, B$ **do**
7:         Break down the prompt-response pair $(\boldsymbol{x}_i, \boldsymbol{y}_i)$ into state-action (prefix-next token) pairs $\{(s_t, a_t)\}_{t=1}^T$.
8:         Query reward function to get $r_t = r(s_t, a_t)$ for $t \in [T]$.
9:         Compute advantages $\hat{A}_t$ using GAE (5) for $t \in [T]$.
10:        Collect batch data $\mathcal{B} \leftarrow \mathcal{B} \cup \{(s_t, a_t, r_t, \hat{A}_t)\}_{t=1}^T$.
11:    **end for**
12:    Update critic parameters $\phi$ by minimizing $\mathcal{L}_{\text{critic}}(\phi)$ defined in (6).
13:    Update actor parameters $\theta$ by minimizing $\mathcal{L}_{\text{actor}}(\theta)$ defined in (7).
14: **end for**
**Output:** Trained policy model $\pi_\theta$.

---

In Algorithm 2, the GAE with hyperparameters $\gamma$ and $\lambda$ is defined as:

$$\hat{A}_t = \sum_{l=0}^{\infty} (\gamma\lambda)^l \delta_{t+l}, \quad \text{where} \quad \delta_t = r_t + \gamma V_\phi(s_{t+1}) - V_\phi(s_t). \tag{5}$$

The critic loss $\mathcal{L}_{\text{critic}}(\phi)$ is defined as:

$$\mathcal{L}_{\text{critic}} = \hat{\mathbb{E}}_{\mathcal{B}} \left[ \left( V_\phi(s_t) - R_t \right)^2 \right], \quad \text{where} \quad R_t = \sum_{k=0}^{\infty} \gamma^k r_{t+k}. \tag{6}$$

The actor loss $\mathcal{L}_{\text{actor}}(\theta)$ is defined as:

$$\mathcal{L}_{\text{actor}}(\theta) = \hat{\mathbb{E}}_{\mathcal{B}} \left[ -\min \left( \frac{\pi_\theta(a_t|s_t)}{\pi_{\theta_{\text{old}}}(a_t|s_t)} \hat{A}_t, \text{clip} \left( \frac{\pi_\theta(a_t|s_t)}{\pi_{\theta_{\text{old}}}(a_t|s_t)}, 1-\epsilon, 1+\epsilon \right) \hat{A}_t \right) \right] \tag{7}$$

where $\pi_{\theta_{\text{old}}}$ is the policy before the update, and $\epsilon$ is the clipping parameter. In our experiments, we set $\epsilon = 0.2$, $\gamma = 1$ and $\lambda = 0.95$.

# C    Related Work

Previous works have explored framing LLM alignment as an imitation learning (IL) problem Sun & van der Schaar (2024); Wulfmeier et al. (2024); Li et al. (2024); Cho (2024). This approach views the maximum likelihood estimation in standard SFT as behavior cloning

(BC) (Pomerleau, 1991), which directly maps states to expert demonstrations. Although straightforward, BC is often unreliable due to challenges with out-of-domain generalization and compounding errors Ross et al. (2011); Chang et al. (2021). To address these limitations, pioneering studies have introduced Inverse Reinforcement Learning (IRL) (Ng et al., 2000; Ziebart et al., 2008) and Adversarial Imitation Learning (AIL) (Ho & Ermon, 2016), which aim to infer a reward function that captures underlying objectives of the expert, leading to more robust policies that generalize better to new scenarios. These methods have become the foundation for contemporary IRL and AIL-based techniques tailored for LLM fine-tuning Chen et al. (2024); Sun & van der Schaar (2024); Wulfmeier et al. (2024); Li et al. (2024). Inspired by recent advancements in Coherent Soft Imitation Learning (CSIL) (Watson et al., 2024), we introduce a combination of BC and IRL approach using a novel coherent reward that measures the divergence between the fine-tuned policy and the pre-trained policy, removing the need to explicitly or implicitly train a separate reward function.

The current state-of-the-art method for aligning large language models (LLMs) begins with SFT using behavior cloning (BC) Ouyang et al. (2022); Wei et al. (2022a) or inverse reinforcement learning (IRL) Sun & van der Schaar (2024); Li et al. (2024); Wulfmeier et al. (2024) on pairs of instructions and demonstrations. Next, RLHF is applied, relying on additional preference-annotated data Christiano et al. (2017); Ziegler et al. (2019); Stiennon et al. (2020); Bai et al. (2022b); Li et al. (2023); Chan et al. (2024). RLHF uses a separate reward model trained on these preferences and optimizes the SFT model using on-policy RL techniques such as PPO (Schulman et al., 2017). Despite the success of these methods, collecting preference annotations is costly. Additionally, issues like overfitting make reward modeling susceptible to overoptimization or reward hacking Skalse et al. (2022); Gao et al. (2023); Guo et al. (2025), which can lead to undesirable behaviors in the target policy. In contrast, our work removes the need for expensive preference annotations and sensitive reward modeling.

The significance of on-policy learning in LLM alignment has been widely discussed Guo et al. (2024); Dong et al. (2024); Tang et al. (2024); Tajwar et al. (2024). Several studies have shown that on-policy methods, such as PPO, outperform off-policy methods like direct preference optimization (DPO) Rafailov et al. (2024a) in terms of out-of-distribution generalization, reasoning capabilities, and generation diversity Xu et al. (2024); Ivison et al. (2024); Tang et al. (2024). Addtionally, on-policy methods avoid the distributional gap between data collection and the target policy, which can lead to suboptimal optimization in off-policy approaches Tajwar et al. (2024); Zhou et al. (2024). Our method combines a coherent reward with PPO to iteratively align large language models. The on-policy nature of PPO ensures that the policy is continuously updated based on its current behavior and allows for the exploration of a diverse response space, improving the model generalization to new scenarios.

## D Hyperparameter Setup

In this section, we present the detailed hyperparameters for experimental results in Tables 1 to 4.

Table 5: Hyper-parameter setup for SFT with TULU-v2-mix and a 9k subset of Ultrafeedback demonstrations.

| SFT Training Data | Model | Method | learning rate or actor learning rate | Epochs or Steps |
|---|---|---|---|---|
| TULU-v2-mix and 9k Ultra-feedback demonstration | Mistral-7B | SFT | $1 \times 10^{-6}$ | 10 epochs |
| | | SRPPO | $5 \times 10^{-8}$ | 3 episodes |
| | LLAMA3-8B | SFT | $5 \times 10^{-6}$ | 10 epochs |
| | | SRPPO | $5 \times 10^{-8}$ | 3 episodes |

Table 6: Hyper-parameter setup.

| SFT Training Data | Model | Method | learning rate | Epochs or Steps |
|---|---|---|---|---|
| TULU-v2-mix | Mistral-7B | SFT | $5 \times 10^{-6}$ | 2 epochs |
| | | SFT (Extended) | $5 \times 10^{-6}$ | 6 epochs |
| | | SPIN (the first iteration) | $5 \times 10^{-7}$ | 2500 steps |
| | | SRPPO | $5 \times 10^{-8}$ | 2 episodes |
| | LLAMA3-8B | SFT | $1 \times 10^{-5}$ | 2 epochs |
| | | SFT (Extended) | $1 \times 10^{-5}$ | 6 epochs |
| | | PPO w/ preference RM | $5 \times 10^{-8}$ | 2 episodes |
| | | SRPPO | $5 \times 10^{-8}$ | 3 episodes |

Table 7: Hyper-parameter setup for SFT with TULU-v2-mix and a 9k subset of Ultrafeedback demonstrations, and 40k subset of TULU-v2-mix.

| SFT Training Data | Model | Method | learning rate or actor learning rate | Epochs or Steps |
|---|---|---|---|---|
| TULU-v2-mix and 9k Ultra-feedback demonstration | Mistral-7B | SFT | $1 \times 10^{-6}$ | 8 epochs |
| | | SRPPO | $5 \times 10^{-8}$ | 2 episodes |

# E    Token-wise Coherent Reward

We assign the coherent reward at the process level as defined in (4). Alternatively, we can revise the process-level coherent reward to a token-wise reward and assign it at token level:

$$r(y_j|\boldsymbol{x}, \boldsymbol{y}_{<j}) = \log \frac{p_{\boldsymbol{\theta}^{(\text{SFT})}}(y_j|\boldsymbol{x}, \boldsymbol{y}_{<j})}{p_{\boldsymbol{\theta}^{(\text{PT})}}(y_j|\boldsymbol{x}, \boldsymbol{y}_{<j})}. \tag{8}$$

However, during training, we observed a significant issue: the model consistently biased toward generating increasingly long sequences. This behavior lead to deteriorated performance, as the model produced unnaturally verbose outputs. Upon further analysis, we identified that the uncontrolled length growth stemmed from the token-wise reward design:

- Almost every generated token was assigned a non-negative reward, regardless of its contribution to the sequence's overall quality.
- The [EOS] (end-of-sequence) token, which signals termination of the generation, did not receive any distinguishable reward signal.

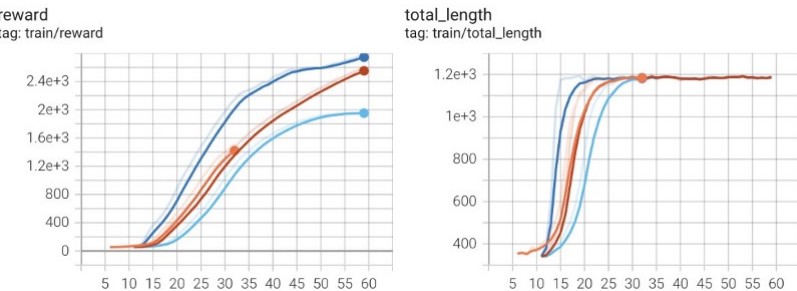

Figure 2: PPO training curve using the token-wise coherent reward (8).

As a result, the model learned to extend sequences unnecessarily, maximizing cumulative rewards without considering the intended stopping point. To address this issue, we restructured the reward formulation to focus on the entire sequence rather than individual tokens:

- We calculated the log policy ratio for the likelihood of the entire sequence.
- This sequence-level reward was then assigned exclusively to the [EOS] token, effectively treating it as a summary evaluation of the entire generation.

By tying the reward to the [EOS] token, we resolved the uncontrolled length growth issue, as the model was no longer incentivized to generate excessively long sequences. Instead, the reward mechanism now encourages the model to generate sequences of appropriate length that align well with the desired policy behavior. This reward redesign successfully stabilized PPO training and mitigated the length bias problem. It ensured that the model balanced the trade-off between sequence quality and length, resulting in outputs that were both coherent and concise. Our solution highlights the importance of carefully designing reward mechanisms in reinforcement learning settings, particularly for autoregressive generation tasks where sequence length plays a critical role.

