# OpenReview forum: "Self-Rewarding PPO: Aligning Large Language Models with Demonstrations Only"
_colmweb.org/COLM/2025/Conference — COLM 2025_

### Official Review · Reviewer_vgGq · 2025-05-13

**Rating:** 6
**Confidence:** 3
**Ethics Flag:** 1

**Summary:**

The paper presents a simple yet effective strategy called Self-Rewarding PPO (SRPPO) that enables fine-tuning a model with only demonstrations (positive examples), doing without preference data to perform reinforcement learning by using a coherent reward based on a log policy ratio between the SFT model and the pretrained model. SRPPO enables on-policy fine-tuning without preference data by addressing the limitations of overfitting of SFT and improves generalization by being able to continue training with unseen prompts. The authors show that SRPPO outperforms or performs on par for various instruction-following and reasoning benchmarks compared to various baselines for Mistral 7B and LLAMA3-8B.

**Questions To Authors:**

- line 77: pretrianed -> pretrained
- line 88: what is meant by mid-aligned? why is a SFT model mid-aligned as opposed to fully aligned?
- Missing related work: https://arxiv.org/abs/2406.00888
- line 170-173 is redundant
- line 196: rewarding learning -> reward learning
- line 236: realted -> related
- line 239-242: could the shift from TULU-v2-mix to UltraFeedback be quantified concretely in order to back up the validity of this setup for measuring generalizability? I know this is somewhat measured by the evaluations that vary the degree of overlap between SFT and PPO data but intuitively I would expect coherent reward to only provide correct reward signals when the SFT data and the prompts seen in PPO are from a similar domain for SRPPO to be effective.
- PPO is generally challenging to work with to find hyperparameters with good performance. Can you share details on this search process? Do you observe large fluctuations in performance depending on the chosen set of hyperparameters?
- Another interesting experiment would be to see how using different epochs of SFT and size of SFT data affects the performance of SRPPO to balance the effect of overfitting to the training distribution while having the SFT Policy functioning as an effective coherent reward. Being able to balance this tradeoff effectively would further add to the usefulness of SRPPO.
- Can you further elaborate on coherent soft imitation learning and how it motivates SRPPO?

**Reasons To Accept:**

- SRPPO is a simple yet effective method that outperforms or performs on par for various benchmarks. The authors also provide results with varying degrees of overlap between SFT and PPO data to demonstrate the effectiveness of SRPPO in generalizing beyond the data that it seen during SFT.
- The paper is well-written, making it easy to follow and understand the main research question presented in lines 153-154. The essence of the paper's contributions is clearly situated in relation to prior related work.

**Reasons To Reject:**

- Improvements for general instruction-following tasks seem clear while those for reasoning and math-related tasks are less evident compared to baselines. (GSM8k and GPQA in table 1 and 2).
- Experiments are limited to relatively small models and therefore it is unclear whether SRPPO is effective for larger models as well.
- The benefit of SRPPO is limited to a specific scenario where high-quality responses are scarce but similar prompts are abundant (lines 91-92) and thus window of opportunity for this method to be practical may be narrow. The authors do not experiment with a systematic variation in the amount of SFT data to establish the data requirements for SRPPO.

---

> ### Author Response · Authors · 2025-06-03
> **Author Rebuttal**
>
> **Q1: Improvement in math and reasoning is less evident.**
>
> We appreciate the reviewer’s observation. In our experiments, the SFT policy is trained on the TULU-v2-mix dataset, which contains limited coverage of math and multi-step reasoning tasks. Consequently, the SFT policy exhibits weaker performance in these domains, and since SRPPO derives its reward signal directly from the SFT model, this limitation naturally affects the effectiveness of downstream performance.
>
> That said, SRPPO still demonstrates consistent gains over the SFT baseline even in math and reasoning tasks—despite no access to preference data or task-specific supervision. We believe this highlights the robustness of our method. Moreover, we expect that incorporating more math-focused demonstrations during the SFT stage would further improve the coherent reward and lead to stronger gains on math and reasoning benchmarks, which we plan to explore in future.
>
> **Q2: Experiments are limited to small models.**
>
> PPO-type methods require hosting multiple models, including actor, reference, and critic models, which consumes a lot of memory and computation. Therefore, Llama-8B is the largest model we can afford given our limited computational resources. Nevertheless, we expect our result to hold for larger models, which can be examined with more resources.
>
> **Q3: The authors do not experiment with a systematic variation in the amount of SFT data to establish the data requirements for SRPPO.**
>
> We appreciate the reviewer’s point and agree that understanding how SRPPO responds to different SFT data regimes is important.  To systematically investigate this, we define three experimental setups in Section 4.1: (1) Minimum overlap, where SFT demonstrations and PPO prompts are drawn from disjoint sources. (2) Medium overlap, where a small subset of PPO prompts are included in the SFT data. (3) Diminished overlap, where we introduce additional training data to reduce overfitting to overlapping prompts.
>
> These setups allow us to examine how different levels of prompt overlap affect the coherence and generalization of the reward, and ultimately, the effectiveness of SRPPO. As shown in Tables 1–3, SRPPO consistently improves performance even under minimal overlap, demonstrating that the coherent reward generalizes beyond the original demonstration data, enabling scalable alignment without requiring additional annotations.
>
>
> **Q4: Typos in line 77, 170-173, 196, 236, and missing related work https://arxiv.org/abs/2406.00888.**
>
> Thank you for catching these typos and mentioning related work. We will fix them in the revision.
>
> **Q5: Why is an SFT model mid-aligned?**
>
> Recent RLHF practices suggest that SFT is not enough for getting a highly aligned model. While SFT can make the model follow some instructions, the generalization is largely limited by the scarce gold responses. Therefore, a typical framework is to leverage human preference data and apply offline (e.g. DPO) or online policy optimization methods (e.g. PPO) after SFT for better alignment. Thus, we call an SFT model mid-aligned as it only performs the first half of the pipeline.

---

> > ### Comment · Reviewer_vgGq · 2025-06-06
> >
> > Thank you for your rebuttal.
> >
> > I agree that the overlap experiments are interesting, but they don't address my point about the variation in amount of SFT data. I am curious as to whether there is a certain amount of SFT demonstrations that are needed in order for SRPPO to work. If we apply SRPPO directly on a pre-trained model, it would likely do poorly, so is there any data requirement of SFT data for SRPPO to work? While I believe this paper makes a meaningful contribution on its own with the current set of experiments, (hence the marginally above acceptance threshold rating), but it would be more complete with this variation for the readers to get a sense of SRPPO's potential limitations for a model that was trained only on a small amount of SFT data for whatever reason (specific downstream task/domain for which data is more scarce).

---

### Official Review · Reviewer_m2aH · 2025-05-13

**Rating:** 6
**Confidence:** 3
**Ethics Flag:** 1

**Summary:**

The authors propose finetuning/align LLMs in a staggered manner: First use supervised finetuning (SFT) on a demnstration dataset, then do reinforcement learning on the target domain data using the ratio between the finetuned model and the baseline model as reward. This way, there's no need for a critic model.
In the experiments shown this approach seems to work well.

**Questions To Authors:**

- Line 3: Can you rewrite this sentence such that it's clearer what "off-policy" means here?
- Line 122: It would be helpful to not just state "clearly" here.
- Eq. 2: Can you note here what $p_{ref}$ is in your case?
- Alg. 1: Typo "Demonstartion"

**Reasons To Accept:**

- Applying RLHF using only one model with good results.

**Reasons To Reject:**

- I'm a bit doubtful about the baselines, specifically there's no other true baseline than a DPO-based approach. A comparison to "Self-Rewarding Language Models" would be helpful.
- Hard to follow at times, lots of RL lingo even in the abstract.

---

> ### Author Response · Authors · 2025-06-03
> **Author Rebuttal**
>
> **Q1: Can you rewrite this sentence such that it's clearer what "off-policy" means here?**
>
> We will clarify that “SFT, being an off-policy method, learns solely from expert-provided demonstrations rather than adapting to its own generated outputs.”
>
> **Q2: I'm a bit doubtful about the baselines, specifically there's no other true baseline than a DPO-based approach. A comparison to "Self-Rewarding Language Models" would be helpful.**
>
> We appreciate the reviewer’s suggestion and would like to clarify the scope and baseline selection in our work. As described in Section 1, the standard LLM alignment pipeline includes: (1) Supervised fine-tuning (SFT) with high-quality (prompt, response) demonstrations, and (2) RLHF with human preference data to train a reward model and further fine-tune the policy.
>
> Our work focuses exclusively on stage (1)—alignment from demonstrations without any preference annotations. Accordingly, our primary baseline is SFT trained on the same demonstration data. Additionally, we include SPIN as a representative DPO-style baseline that also aligns from demonstrations and does not require preference data, making it a directly comparable method.
>
> To broaden context, we also include a PPO baseline with a simple reward model (Fsfairx-LLAMA3-RM) in Table 2. While stronger reward models (e.g., UltraRM, DeepSeek-RM) could improve RLHF results, our aim is to show that SRPPO—despite not using preference data—can already match or outperform these preference-based methods under the same annotation budget.
>
> Regarding the “Self-Rewarding Language Models (SRLM)” paper, we thank the reviewer for the suggestion. SRLM proposes a contrastive learning setup built on pairwise comparisons generated from a model’s own outputs. In contrast, our approach derives a reward signal from the log-likelihood ratio between the SFT and pretrained models, which aligns more directly with demonstration-based supervision and is compatible with smaller models.
>
> Moreover, SRLM is demonstrated primarily on 70B-scale models in open-ended generation tasks, and requires substantially more compute and sampling. SRPPO, by contrast, is lightweight, scalable to smaller models, and optimized for efficiency in the low-data regime. We will clarify these distinctions in the revision and add discussion of SRLM in the related work section.

---

> > ### Comment · Reviewer_m2aH · 2025-06-08
> >
> > Thank you for your answers, which, also considering the other reviews, strengthen my current assessment.

---

### Official Review · Reviewer_zG4C · 2025-05-23

**Rating:** 5
**Confidence:** 3
**Ethics Flag:** 1

**Summary:**

This paper introduces self-rewarding PPO (SRPPO), a fine-tuning method that combines supervised fine-tuning (SFT) and proximal policy optimization (PPO), without the need for extra human preference labels or reward models. Specifically, SRPPO begins with an SFT-ed policy, which is then optimized using PPO with a coherent reward, inspired by coherent soft imitation learning. The coherent reward is defined as the log ratio between the SFT-ed policy and the pre-trained base model. Experiments on Mistral-7B and Llama3-8B demonstrate improvements over SFT-only and other baselines.

**Reasons To Accept:**

- The motivation and the method sections are well written and easy to follow.
- SRPPO eliminates the need for human preferences, thus it is more cost effective than other alignment methods.
- Experimental results show promising gains of two open-source models across a variety of tasks.

**Reasons To Reject:**

The paper can be greatly improved with (1) additional stronger and more controlled baselines, and (2) additional analysis on how the quality of SFT-ed policy impacts SRPPO.

If I read it correctly, the extended-SFT baseline does not leverage the exactly same prompt distribution used by SRPPO stage. A stronger SFT baseline would be continue the SFT model (initial policy for SRPPO RL finetuning) with demonstrations where the prompt is used by SRPPO stage for all three data overlap settings. With current results, it is unclear if the improvement is by data or by better algorithmic design. Similarly for the SPIN baseline, making sure the prompt distribution is the same can help strengthen the results.

The reward model baseline experiment can be further improved. The PPO baseline was using an external reward model without reasons for why that particular RM was chosen. There are also missing values for the PPO baseline row in the table. Trying a few more reward models and showing consistent gains over those PPO baselines can help increase the confidence on the performance of SRPPO.

The proposed coherent reward (log p_SFT / p_PT) seems to be only meaningful when the SFT-ed policy is noticeably better than the base model. More (theoretical or empirical) analysis on how the quality of the SFT policy influences the SRPPO tuning can help better understand the benefits and the limitations of the proposed method.

---

> ### Author Response · Authors · 2025-06-03
> **Author Rebuttal**
>
> **Q1: A stronger SFT baseline would continue the SFT model (initial policy for SRPPO RL finetuning) with demonstrations where the prompt is used by SRPPO stage for all three data overlap settings.**
>
> We appreciate the reviewer’s suggestion and agree that aligning the prompt distributions across baselines can offer a more controlled comparison. However, we would like to clarify a core motivation behind our method. In practice, prompts without paired responses are far easier to obtain at scale, while collecting high-quality demonstrations (i.e., expert-written responses) is non-trivial and costly. Methods like SFT and SPIN require human-annotated responses to train effectively on new prompts, which incurs significant labor and annotation overhead. In contrast, a key contribution of SRPPO is that it can leverage additional prompts without requiring any new ground-truth responses. By deriving a coherent reward from the SFT policy itself, SRPPO enables on-policy fine-tuning solely from unlabeled prompts, effectively amplifying the utility of a small, fixed set of high-quality demonstrations.
>
> In our current experimental setup, all methods are trained using the same amount of human-annotated responses to ensure a fair comparison in terms of supervision cost. Introducing additional labeled responses for SFT or SPIN to match the prompt distribution used by SRPPO would result in unequal supervision budgets, making the comparison less meaningful. We agree it is meaningful to assess whether SRPPO’s improvement stems from broader prompt coverage or the algorithm itself. To clarify, it is precisely our algorithmic design that allows SRPPO to effectively utilize broader prompt distributions—without requiring additional annotations—to enhance LLM alignment performance.
>
>
> **Q2: The reward model baseline experiment can be further improved.**
>
> We appreciate the reviewer’s suggestion and would like to clarify the scope and intent of our work. As outlined in Section 1, the standard LLM alignment pipeline typically consists of two stages: (1) Supervised fine-tuning (SFT) using demonstration data—i.e., high-quality (prompt, response) pairs annotated by experts; and (2) RLHF via preference learning, which requires separate human-annotated comparison data to train a reward model and fine-tune the policy through reinforcement learning.
>
> In this paper, we focus exclusively on the first stage—alignment from demonstrations without preference annotations. Consequently, we do not use any human-labeled preference data, and RLHF methods that rely on such data fall outside the scope of our paper.  Our primary baseline is therefore SFT using the same demonstration data. Nevertheless, in Table 2, we included results from RLHF with a basic preference-based reward model (Fsfairx-LLAMA3-RM) as an additional point of reference. While more advanced reward models (e.g., UltraRM, DeepSeek-RM) could indeed yield higher RLHF performance, our intention was to illustrate that SRPPO, despite using no preference data, can already match or outperform basic RLHF pipelines.
>
> **Q3: The proposed coherent reward (log p_SFT / p_PT) seems to be only meaningful when the SFT-ed policy is noticeably better than the base model.**
>
>
> We agree with the reviewer that the effectiveness of the coherent reward depends on the quality of the SFT policy. Intuitively, if the SFT policy fails to meaningfully improve over the pretrained baseline—e.g., due to low-quality or misaligned demonstration data—then the resulting reward signal may be weak and unable to provide meaningful signal. Indeed, the quality of SFT plays a crucial role in establishing a useful alignment direction, which SRPPO builds upon through on-policy fine-tuning. Ensuring that the SFT model captures meaningful improvements over the base model is therefore essential for the coherent reward to provide reliable training signals during the PPO stage.

---

> > ### Comment · Reviewer_zG4C · 2025-06-07
> >
> > Thank you for the detailed response! I understand that one of the benefits of the proposed method is to bypass human labeled preference and also mentioned that as one of the reasons to accept. Still having controlled data distribution for baselines and the proposed method would help understand the method better. And showing more results that the proposed method can robustly work for SFT policies obtained from various data distributions would also greatly strengthen the paper. This would also directly address the Q3, regarding the analysis on the quality of SFT policy. The proposed method is very clean, simple and easy to implement, and it would be great to see more empirical results with controlled data setting and beyond current relatively constrained data selection!

---

### Official Review · Reviewer_fYjb · 2025-05-24

**Rating:** 6
**Confidence:** 3
**Ethics Flag:** 1

**Summary:**

This paper introduces an RLHF approach that does not rely on preference data but only high-quality demonstration data. Specifically, the authors compute the logarithm of the relative ratio of the SFT model and the base model response probabilities to derive a reward that reflects how well the response aligns with the demonstration data. Then the authors apply PPO to further optimize the reward. The proposed method is evaluated with Mistral-7B and LLaMA3-8B on four math&chat benchmarks.

**Questions To Authors:**

1. Why could the proposed method be better than PPO with a preference reward model?
2. How would the quality of demonstration data influence the performance of SRPPO?

**Reasons To Accept:**

1. This paper introduces an RLHF method that does not rely on preference data but only high-quality demonstration data.
2. The authors ablate several choices on the SFT data, demonstrating the necessity to adopt a minimum overlap setup for the SFT and PPO stages.
3. The authors draw a connection between the proposed method and Coherent Soft Imitation Learning.

**Reasons To Reject:**

1. Lack of analysis on why the proposed method could be better than PPO with a preference reward model. It is expected that the authors provide a theoretical analysis on how to frame SRPPO within the framework of inverse RL or Coherent Soft Imitation Learning.
2. Lack of an ablation study on the quality of the demonstration data. Though being free of the preference dataset makes the method promising, the quality of demonstration data should be an important influencing factor for SRPPO.
3. There are limited baselines. It would be better to incorporate more baselines such as DPO and model merging.

---

> ### Author Response · Authors · 2025-06-03
> **Author Rebuttal**
>
> **Q1: Why could the proposed method be better than PPO with a preference reward model?**
>
> Thank you for the insightful question. One of the key advantages of our proposed method lies in the coherent reward formulation, which is derived directly from the SFT policy rather than a separately trained reward model. it is inherently sensitive to variations in the model’s own responses $y\sim p_{\theta}$ during on-policy RL fine-tuning. In contrast, PPO with an external preference reward model often depends on human-labeled comparison data, which can be noisy or inconsistent. Moreover, such reward models may fail to generalize well beyond the preference distribution they were trained on—especially when applied to specific tasks or unseen domains. As shown in Table 2, SRPPO can match or outperform PPO with a basic public reward model, despite requiring less human supervision.
>
> **Q2: How would the quality of demonstration data influence the performance of SRPPO?**
>
> Since the coherent reward in SRPPO is derived from the SFT policy, the quality and coverage of the demonstration data used for SFT are critical to the effectiveness of the reward and downstream fine-tuning. A well-trained SFT policy—grounded in high-quality, diverse demonstrations—provides a more meaningful and stable reward signal for on-policy learning.
>
> Empirically, we observe that greater overlap between SFT training data and PPO prompts leads to more robust reward evaluations and improved performance. To systematically study this, we design three experimental setups (Section 4.1): (1) Minimum overlap, testing generalization to unseen prompts; (2) Medium overlap, simulating partial prompt coverage; (3) Diminished overlap, where overlapping prompts are diluted by unrelated ones.
>
> As shown in Tables 1–3, SRPPO demonstrates strong generalization even under limited overlap, confirming that it can effectively leverage additional unlabeled prompts to improve performance—without requiring human preferences.
>
> In future revisions, we will include additional ablation studies on demonstration quality, such as: (1) Varying the number and diversity of demonstration samples. (2) Testing on different datasets to evaluate how SRPPO generalizes across domains. (3) Comparing SRPPO performance under synthetic, low-quality demonstrations vs. expert-curated ones.

---

> ### Comment · Reviewer_mRvM · 2025-06-10
> **Rebuttal**
>
> I have read the response from the authors.  While I appreciate their efforts, I think the lack of SPIN results in addition to the other factors means that the paper probably should go through another round of revision before final submission.  I will hold my rating.

---

### Official Review · Reviewer_mRvM · 2025-05-27

**Rating:** 4
**Confidence:** 4
**Ethics Flag:** 1

**Summary:**

SFT is an off policy approach similar to behavior cloning and suffers from overfitting or poor out of domain generalization.  SRPPO is claimed to be a novel fine tuning method that leverages on policy techniques to enhance generalization.  The reward function is a log likelihood ratio between the SFT model and the base model in this paper (e.g., self-rewarding).  This enables no preference annotations.  The empirical evaluation is claimed to demonstrate outperformance of SFT methods.

The intro goes into SFT and preference learning for alignment.  RLHF suffers from challenges in deriving a reward signal.  Alternative approaches include IRL. SPIN bypasses this by applying DPO.  The authors call this a “coherent reward.”  The authors apply PPO to continuously fine-tune the model using a set of prompts.  The paper was inspired by coherent soft imitation learning.  Compared to SPIN, SRPPO allows the use of additional prompts.  Section 2 gives the background, including SFT and RLHF.  Section 3 gives the approach, including the coherent reward definition and self-rewarding PPO.  Section 4 gives results.

**Questions To Authors:**

How much SFT should be done is never discussed in the paper.  The whole aim here is to only do a bit of SFT, I would assume and use that directionally for downstream performance.

It’s also not clear if there was ever a full comparison to SFT on downstream tasks.

* It was mentioned multiple places in the paper that compared to SPIN, you get additional prompts.  What does this buy you? Why was SPIN not evaluated in Table 2, Table 3 or Table 4?  SPIN / DPO should generally be evaluated.
* Why don't you compare against approaches that have human feedback?  It's not clear how, with these approaches, you get a resulting model that is resonably aligned with human preferences without human feedback?
* I wanted there to be a figure like Figure 2 in the SPIN paper, but with SRPPO.  Were the same metrics evaluated as in that paper?
* The reasons for the three setups on page 7 was never intuitively explained.  And diminished and minimum on the surface have similar meanings.  What was the reason for each experiment?  Why were multiple SFT subsets not evaluated?
* It would be much more impactful to release the code.  Is that possible


Small things:
P2, pretrianed -> pretrained
P3, pretarined -> pretrained

**Reasons To Accept:**

Growing a weak LLM into strong one without human feedback is a promising and interesting research direction.  The authors present an approach toward that goal.

**Reasons To Reject:**

There are a number of apparent issues with the paper that put the results into question:
* SPIN (a reasonable baseline) is not evaluated across all scenarios.
* There is no apparent baseline with human feedback.
* For some decisions (e.g., reward function) and experiments (the three experimental setups on page 7), it was not clear why these were made or what intuitions we should derive from them.  See questions below.
* The code does not appear to be released, which will limit reproducibility.
* The differences from SPIN appear to be minimal and it is not clear the experiments are as complete as in that paper, despite substantial overlap in approach.

---

> ### Author Response · Authors · 2025-06-03
> **Author Rebuttal**
>
> **Q1: There is no apparent baseline with human feedback.**
>
> We appreciate the reviewer’s question and would like to clarify the scope of our work. As discussed in Section 1, the alignment process for large language models typically consists of two stages: (1) Supervised fine-tuning (SFT) using demonstration data—i.e., high-quality (prompt, response) pairs labeled by experts; and (2) Reinforcement learning from human feedback (RLHF), which uses human-annotated preference data to train a reward model and further fine-tune the policy.
>
> In this paper, we focus exclusively on the first stage—alignment from demonstrations without preference annotations. Consequently, we do not use any human-labeled preference data, and RLHF methods that rely on such data fall outside the scope of our paper.  Our primary baseline is therefore SFT using the same demonstration data. Nevertheless, in Table 2, we included results from RLHF with a basic preference-based reward model (Fsfairx-LLAMA3-RM) as an additional point of reference. While stronger reward models (e.g., UltraRM, DeepSeek-RM) could improve RLHF performance, our goal is to demonstrate that SRPPO—despite relying only on demonstration data—can already match or exceed such basic preference-based approaches.
>
> **Q2: The differences from SPIN appear to be minimal. The overlap in approach is substantial.**
>
> We appreciate the reviewer’s question and would like to clarify the key distinctions between SRPPO and SPIN. As discussed in Section 1, SPIN relies on the assumption that demonstration responses are always preferred over model-sampled alternatives, and optimizes a DPO-style objective using only labeled (prompt, response) pairs. Consequently, SPIN is restricted to training on demonstration data and cannot make use of unlabeled prompts—a limitation in settings where high-quality responses are scarce and expensive to annotate.
>
> In contrast, our method SRPPO derives a coherent reward function explicitly from SFT and pretrained policies. This self-derived reward serves as a training signal for a range of on-policy reinforcement learning algorithms (e.g., PPO, REINFORCE, RLOO). A core advantage of our approach is that it enables on-policy fine-tuning using additional prompts without requiring new human-labeled responses. This allows SRPPO to generalize alignment beyond the limited coverage of the demonstration data.
>
> While both methods align from demonstrations without preference labels, SRPPO’s ability to perform on-policy sampling and incorporate unlabeled prompts is a fundamental difference. This flexibility is particularly important in practice, where high-quality responses are expensive to annotate, but diverse prompts are easy to collect.
>
> **Q3: For some decisions (e.g., reward function) and experiments (the three experimental setups on page 7), it was not clear why these were made or what intuitions we should derive from them.**
>
> We appreciate the reviewer’s question and would like to clarify the rationale behind both our reward function and the three experimental setups. As discussed in Q2, a key advantage of SRPPO is its ability to leverage additional unlabeled prompts during fine-tuning—thanks to the coherent reward, which is derived directly from the SFT and pretrained policies. This makes SRPPO especially valuable in settings where high-quality responses are limited, but prompts are abundant.
>
> However, since the reward signal is computed from the SFT policy, the choice of SFT training data directly impacts the effectiveness of generalization to new prompts. Empirically, we observe that greater overlap between SFT demonstrations and PPO prompts leads to stronger and more stable coherent rewards.
>
> To systematically investigate this, we define three experimental setups in Section 4.1: (1) Minimum overlap, where SFT demonstrations and PPO prompts are drawn from disjoint sources. (2) Medium overlap, where a small subset of PPO prompts are included in the SFT data. (3) Diminished overlap, where we introduce additional training data to reduce overfitting to overlapping prompts.
>
> These setups allow us to examine how different levels of prompt overlap affect the coherence and generalization of the reward, and ultimately, the effectiveness of SRPPO. As shown in Tables 1–3, SRPPO consistently improves performance even under minimal overlap, demonstrating that the coherent reward generalizes beyond the original demonstration data, enabling scalable alignment without requiring additional annotations.
>
> **Q4: SPIN (a reasonable baseline) is not evaluated across all scenarios.**
>
> In the revised version, we will add SPIN results for the medium and diminished overlap settings, using the subset of prompts for which expert responses are available. This will allow for a more complete baseline comparison while respecting the supervision requirements of SPIN.
>
> **Q5: Is it possible to release the code.**
>
> Our code will be publicly available when we release the paper.

---

### Decision · Program_Chairs · 2025-07-08

**Decision:**

Accept

**Comment:**

The paper proposes SRPPO, a method for instruction-tuning from demonstrations without human preference data. The main idea is to use a _coherent reward_, which is the log-ratio of the sequence likelihood under the SFT'd model to the likelihood under the base model. Basically, you get more reward if your sequence is relatively more likely under the SFT'd model compared to the base model. This reward can be computed on arbitrary sequences, which allows the authors to train on additional prompts, without demonstrations. The authors compare to baselines including SFT, SPIN, RLHF with a simple reward model, and show promising results.

The proposed method is simple and intuitive, and the results are promising. The reviewers raised concerns with the quality of baselines. The method does allow for training on extra data (new prompts), which the baselines can not do. Also, the method does not rely on human preferences, so comparing to RLHF is challenging. As a result, coming up with a fair evaluation setting is somewhat challenging. The authors suggested that they will include additional results for the SPIN method in the updated manuscript.

The authors provide an ablation on the overlap between the demonstrations and additional prompt distribution. Some reviewers suggested that more ablations should be done on the amount and quality of SFT data.

Overall, the paper proposes a method that enables on-policy RL without human preference data, which is a unique property, and could be useful in practice. Concurrent work [1, 2] works in a similar setting, although with a focus on reasoning.

My main concern with the paper is that it is not clear why the proposed coherent reward is a good reward function, and what are the limits of its applicability. With enough optimization, PPO should learn "reward hacks", i.e. idiosyncrasies of the SFT / Base model which allow the policy to get high reward in a way that is not aligned with the true goal. It would be good to see more analysis on the quality of the reward signal, and possibly comparisons to alternative choices, e.g. just the likelihood under the SFT model.

[1] Learning to Reason without External Rewards
[2] Absolute Zero: Reinforced Self-play Reasoning with Zero Data

[Automatically added comment] At least one review was discounted during the decision process due to quality]